# Cobalt Element Effect of Ternary Mesoporous Cerium Lanthanum Solid Solution for the Catalytic Conversion of Methanol and CO_2_ into Dimethyl Carbonate

**DOI:** 10.3390/molecules27010270

**Published:** 2022-01-02

**Authors:** Xu Li, Hua Shi, Yunhan Gu, Qingyan Cheng, Yanji Wang

**Affiliations:** 1Key Laboratory of Green Chemical Technology and High Efficient Energy Saving of Hebei Province, Tianjin Key Laboratory of Chemical Process Safety, School of Chemical Engineering, Hebei University of Technology, Tianjin 300401, China; 201931505068@stu.hebut.edu.cn (X.L.); 202021503008@stu.hebut.edu.cn (Y.G.); yjwang@hebut.edu.cn (Y.W.); 2China Tianchen Engineering Corporation, Tianjin 300499, China

**Keywords:** solid solution, catalyst, cobalt, dimethyl carbonate, oxygen vacancy

## Abstract

A citric acid ligand assisted self-assembly method is used for the synthesis of ternary mesoporous cerium lanthanum solid solution doped with metal elements (Co, Zr, Mg). Their textural property was characterized by X-ray diffraction, transmission electron microscopy, N_2_ adsorption-desorption, X-ray photoelectron spectroscopy and TPD techniques, and so on. The results of catalytic testing for synthesis of dimethyl carbonate (DMC) from CH_3_OH and CO_2_ indicated that the DMC yield reached 316 mmol/g on Ce-La-Co solid solution when the reaction temperature was 413 K and the reaction pressure was 8.0 MPa. It was found that Co had synergistic effect with La and Ce, doping of Co on the mesoporous Ce-La solid solution was helpful to increase the surface area of the catalyst, promote CO_2_ adsorption and activation, and improve the redox performance of solid solution catalyst. The conversion of Co^2+^ to Co^3+^ resulted in the continuous redox cycle between Ce^4+^ and Ce^3+^, and the oxygen vacancy content of the catalyst was increased. Studies have shown that the catalytic performance of Ce-La-Co solid solution is positively correlated with oxygen vacancy content. On this basis, the reaction mechanism of DMC synthesis from CO_2_ and CH_3_OH on the catalyst was speculated.

## 1. Introduction

Dimethyl carbonate (DMC) is an important green chemical intermediate that is widely used in electronic chemicals, medicine, dyes, synthetic materials, and other fields [1,2], known as the “new base” of organic synthesis in the 21st century [3]. In 2020, China promoted the policy of carbon neutralization and carbon peak, solved the “double carbon” problem from a global perspective and at the height of building ecological civilization, putting forward new requirements for green sustainable development. Up to now, domestic DMC production capacity has exceeded one million tons per year and is still in a growing trend. Among the synthetic methods of DMC, the direct synthesis of DMC from CH_3_OH and CO_2_ is a promising pathway, because CO_2_ conforms the principles of “green chemistry” and the high atom efficiency, water is the only by-product. However, CO_2_ is a fully oxidized, thermodynamically stable, and chemically inert molecule, so how to activate CO_2_ is a great challenge. To solve these difficulties, different catalysts have been applied for this reaction to activate CO_2_ and improve the yield of DMC. Compared with homogeneous catalysts, heterogeneous catalysts have been widely studied due to its easy separation and low toxicity. Metal oxides are ideal catalysts for the synthesis of DMC from CO_2_ and CH_3_OH because of simple preparation methods and high selectivity of DMC. Among them, CeO_2_ has become a research hotspot in recent years.

CeO_2_, a cubic fluorite structure oxide, is widely used in catalysts, solid oxide fuel cells, oxygen sensors and sewage treatment [4,5,6,7] due to its remarkable redox ability and abundant oxygen vacancy. Compared with bulk CeO_2_, mesoporous CeO_2_ has abounds and ordered pore structure, which has higher specific surface area, so it can effectively improve performance in practical application. However, pure CeO_2_ has many defects in application, such as poor redox performance at low temperature and easy sintering. Therefore, the current research is to combine CeO_2_ with other metal elements to form solid solutions in order to obtain better properties. The doping of metal ions (La [8,9,10], Zn [11], Mg [12], Zr and Y [13]) on CeO_2_ to replace some Ce ions leads to structural defects and certain amount of oxygen vacancies on its surface. It is found that the oxygen vacancy on CeO_2_ surface can enhance the adsorption and activation of CO_2_, which is beneficial to improve the reactivity of CO_2_ and CH_3_OH to DMC [14]. Adding appropriate amount of metal ion on CeO_2_ can improve the surface acidity and oxygen vacancy content [15]. The doping of metal ions into CeO_2_ lattice can replace part of Ce^4+^ in the lattice, resulting in charge loss, promoting the formation of oxygen vacancies, thereby enhancing the absorption and release of oxygen in solid solutions, and show better catalytic activity.

Atribak and Buenolopez [16] prepared a series of Ce-Zr composite oxides by coprecipitation method and studied their catalytic properties. It was found that with the calcination temperature increase, the thermal stability specific surface area and the catalytic performance decreased. However, the citric acid complexation method can prepare nano-sized oxide grains at lower calcination temperature, which helps to increase the specific surface area of the catalyst. It was found that doping La as structural aids on CeO_2_ to form Ce-La solid solution could increase the desorption of oxygen vacancies and lattice oxygen, which improved the redox performance of the catalyst [17].

In this experiment, ternary mesoporous cerium lanthanum solid solution with high specific surface area was synthesized by ligand assisted self-assembly method. In the presence of 2-cyanopyridine, the effect of metal ions doping on surface oxygen vacancy of Ce-La solid solution was investigated, which are directly related to the resulting reaction performance.

## 2. Results and Discussion

### 2.1. XRD Analysis

The XRD results are shown in Figure 1, which revealed the structural properties of Ce-La-M (M is Co, Zr, Mg) solid solution. Cerium lanthanum solid solution retains the cubic fluorite structure of pure cerium oxide (JCPDS34-039-4). The Ce-La-M exhibited different diffraction peaks at 28.6°, 33.0°, 47.1°, 56.7°, 59.1°, and 69.3° were assigned to (111), (200), (220), (311), (222), and (400), which showed similar diffraction peaks to pure CeO_2_ (PDF#34-039436) [18], which indicated that a good ternary solid solution was formed. Figure 1 also showed that Ce-La-Co appears with weak peaks at the positions of 2θ of 32.88° and 58.96°. The positions of these diffraction angles are consistent with the three strong peaks (32.58°, 47.48°, 58.98°) of LaCoO_3_ perovskite, and it may be a trace amount of LaCoO_3_ perovskite [19]. Comparing the diffraction peaks of Ce-La solid solution before doping, it was found that the diffraction peaks (111) and (200) of partial crystal planes of the prepared solid solution samples shifted to high angles, indicated M^n+^ ions were embedded in the fluorite structure of Ce-La solid solution [20].

The lattice parameters and interplanar spacing of Ce-La-M solid solution were analyzed by Jade. The results are listed in Table 1. It is shown that the lattice parameters and interplanar spacing of (111) crystal plane of Ce-La-M catalysts decreased when M^n+^ entering the lattice of Ce-La solid solution, which is due to that the ionic radiuses of Mg^2+^ (0.72 Å), Zr^4+^ (0.84 Å), Co^2+^ (0.75 Å) are smaller than those of Ce^4+^ (0.87 Å) and La^3+^ (1.03 Å), which caused the lattice contraction of Ce-La-M solid solution [21,22,23]. This also proves that doped metal element enter the lattice of Ce-La to form solid solution.

### 2.2. TEM Images

TEM images of the prepared solid solution are shown in Figure 2. Ce-La solid solution doped with different metal elements exhibited irregular nanoparticles. Ce-La solid solution presented high particle dispersion and regular particles. Ce-La-Zr solid solution displayed a certain degree of agglomeration. Ce-La-Co and Ce-La-Mg solid solution showed low particle dispersion, small particle diameter, and dense particle distribution.

The particle size distribution and average particle size of nanoparticles were analyzed through the statistical measurement of nanostructures. The results showed that the particle size distribution of the solid solution is between 10 and 12 nm, and Ce-La, Ce-La-Co, Ce-La-Zr, Ce-La-Mg are 10.1, 11.2, 10.8, and 11.0 nm, respectively. The result indicated that doped metal elements on Ce-La solid solution did not significantly affect the particle size and particle size distribution.

HRTEM images have identified the crystallographic features of the Ce-La solid solution as shown in Figure 2. All solid solutions had regular crystal structure. The lattice fringes of 0.31 and 0.27 nm are attributed to the (111) and (200) planes of CeO_2_ (JCPDS 34-0394 [24]), respectively. In addition, the lattice fringe spacing (0.19 nm) corresponding to the (110) crystal plane of CeO_2_ was not found, which may be due to the conversion of exposed crystal plane to (111) crystal plane of CeO_2_ annealed at high temperature. The main reason is that among all the crystal planes of CeO_2_, (111) crystal plane has the lowest formation energy and the most stable structure, and the number of (111) crystal planes exposed is the largest among all crystal planes [25], which is consistent with the XRD characterization results. The lattice fringes of M_x_O_y_ were not found in the HRTEM images, indicating that M_x_O_y_ entered the CeO_2_ lattice to form a solid solution [26,27].

### 2.3. N_2_ Adsorption-Desorption

The N_2_ adsorption-desorption isotherms of the Ce-La-M solid solution are shown in Figure 3. All the isotherms in the range of P/P_0_ = 0.5~1.0 exhibited a similar type IV shape with a H2 hysteresis loop, revealing typical of mesoporous nature of the materials. The Brunauer–Emmett–Teller (BET) surface area of all the Ce-La catalysts were summarized in Table 2. It can be seen the specific surface areas of the solid solution catalyst Ce-La-M is about 70 m^2^/g. The pore volume and pore size of catalyst increased slightly after M doping. However, M doping does not significantly affect the texture properties of the Ce-La solid solution. Combined with TEM images, the texture properties of catalysts changed little by doping metal elements.

### 2.4. CO_2_-TPD

Figure 4 shows the nature of the solid solution surface by CO_2_-TPD. It was found that there were the same adsorption sites on the surface of Ce-La solid solution doped with different metal elements. The CO_2_ desorption peaks appeared in the range of 323–473 K, 473–673 K, and 671–873 K, which can be assigned to weak, medium-strength, and strong adsorption, respectively [28]. The weak base sites mainly come from bicarbonate formed by the combination of CO_2_ and hydroxyl groups on the solid solution surface. The medium-strength base sites are mainly derived from the carbonate species in the double-tooth adsorption state and the bridge adsorption state formed by CO_2_ and Ce-O in the solid solution. The formation of strong base sites is mainly due to the formation of monodentate carbonate species by the low coordination between CO_2_ and O^2−^ on the solid solution surface [29].

The total CO_2_ adsorption capacity of different solid solution were 0.66 mmol/g (Ce-La), 0.79 mmol/g (Ce-La-Co), 0.70 mmol/g (Ce-La-Zr), and 0.75 mmol/g (Ce-La-Mg), which are displayed in Table 3. According to the measured CO_2_ adsorption amount, after doped with metal elements, the CO_2_ adsorption amount on the Ce-La-M surface changed, and the CO_2_ adsorption amount on the Ce-La-M surface is increased compared with that on the Ce-La surface. The CO_2_ adsorption amount on the solid solution surface was in the order of Ce-La-Co > Ce-La-Mg > Ce-La-Zr > Ce-La, and the CO_2_ adsorption amount of Ce-La-Co solid solution was higher than that of other solid solutions.

It was found from Table 3 that after Ce-La solid solution was doped with other elements, the number of weak adsorption sites on the surface of Ce-La-Zr and Ce-La-Mg solid solution decreased, and the number of medium-strength adsorption sites on the surface of solid solution increased. The number of strong base sites on the surface of Ce-La-Co solid solution was slightly lower than that on the surface of Ce-La-Mg and Ce-La-Zr solid solution, and the number of strong base sites on the surface of Ce-La-Mg solid solution was the highest.

### 2.5. Concentration of the Surface Oxygen Vacancies and Chemical States of the Catalysts

Figure 5a is the XPS broad spectra of Ce-La solid solution doped with different metal elements. It can be seen from Figure 5a that each solid solution showed the peaks of C 1s (~284.8 eV), Ce 3d (880.0~920.0 eV), La 3d (830~860 eV), and O 1s (519.0~540 eV). Compared with the characteristic peaks of Ce^4+^ and Ce^3+^ in Ce 3d spectra of Figure 5c, the characteristic peaks of Ce^4+^ and Ce^3+^ in Ce-La-M shift to the large binding energy, and the doping of metal increases the binding energy of Ce^4+^ and Ce^3+^ in Ce-La-M. It indicated that a special solid solution structure is formed between doped metal and Ce-La solid solution.

Figure 5c shows the XPS spectra of Ce 3d. According to the literature [30], the Ce 3d spectra can be deconvoluted into eight peaks: U‴ (∼916.8 eV), U″ (∼907.4 eV), U′ (∼903.4 eV), U (∼900.9 eV) represent the spin-orbit splitting peaks of Ce 3d_3/2_, and V‴ (∼898.3 eV), V″ (∼888.9 eV), V′ (∼884.9 eV), and V (∼882.4 eV) represent the spin-orbit splitting peaks of Ce 3d_5/2_. Six peaks of U‴, U″, U, V‴, V″, and V represent the electronic state of Ce^4+^ at 3d^10^4f^0^, and U′ and V′ represent the electronic state of Ce^3+^ at 3d^10^4f^1^. By calculating the ratio of the area of V′ and U′ peaks to the total area of V, V′, V″, V‴, U, U′, U″and U‴ peaks, the concentration ratio of Ce^3+^ can be quantitatively estimated base the following equation. The results are listed in Table 4.
$$C\;[\mathrm{Ce}3^{+}]\;\%=\left(\frac{A\;Ce^{3+}}{A\;Ce^{3+}+A\;Ce^{4+}}\right)$$

ACe^3+^ = A_v′_ + A_U′_

ACe^4+^ = A_v_ + A_v″_ + A_v‴_ + A_U_+ A_U″_+ A_U‴_

ACe^3+^ and ACe^4+^ donate Photoelectron peaks areas of Ce^3+^ and Ce^4+^

It can be seen from Table 4 that with the doping of metal elements, the content of Ce^3+^ on the solid solution surface is increased compared with that of Ce-La solid solution. The order of Ce^3+^ content on the solid solution surface is Ce-La-Co > Ce-La-Zr > Ce-La-Mg > Ce-La, which indicates that the doping of metal elements in the solid solution can promote the partial reduction of Ce^4+^ on the catalyst surface. Combined with XRD characterization, due to the M^n+^ (Zr^4+^ 0.84 Å, Co^2+^ 0.75 Å, and Mg^2+^ 0.72 Å) is smaller than that of Ce^4+^ (0.97 Å). The substitution of M^n+^ for some Ce ions in the crystal structure of CeO_2_ will lead to the lattice contraction of CeO_2_. When this occurs, Ce^4+^ (0.87 Å) with smaller ionic radius spontaneously transforms to Ce^3+^ (1.1 Å) with larger ion radius to compensate for the lattice contraction, resulting in the increase of Ce^3+^ ratio [31].

The O 1s spectra of the solid solution in Figure 5b showed three peaks: The peak at the binding energy of 529.0 eV is attributed to the lattice oxygen (OL); The peak at the binding energy of 530.5 eV is attributed to the oxygen vacancy (OV); The peak at the binding energy of 532.5 eV is attributed to chemisorbed oxygen (OC) [32]. The concentration of OV on the catalyst surface can be estimated by the integrated peak areas base the following equation. The results are listed in Table 4.
$$C\;[O_{V}\%]=\left(\frac{A\;O_{V}}{A\;O_{L}+A\;O_{C}+A\;O_{V}}\right)\text{×}100\%$$

AO_V_, AO_L_ and AO_C_ stand for the photoelectron peak areas of surface oxygen vacancies, chemisorbed and lattice oxygens.

The content of surface oxygen vacancies of solid solutions was estimated according to the area of each peak fitted by the O 1s spectra. The results are shown in Table 4. The content of oxygen vacancies on the solid solutions surface was increased after doping with metal elements, and the order of the content of oxygen vacancies on the catalyst surface was Ce-La-Co > Ce-La-Zr > Ce-La-Mg > Ce-La. The change trend was consistent with the change of Ce^3+^ content, which also indicated that increasing the Ce^3+^ content on the solid solution surface could lead to more surface oxygen vacancies on the solid solution. In general, the chemical valence of Ce ions is 3 or 4, while the valence state of M ions is low. Therefore, some vacancies are generated during substitution to maintain the charge neutrality of ionic crystals and these vacancies are conducive to heterogeneous catalysis [33,34]. Combined with CO_2_-TPD, after doping with metal elements, the change trend of CO_2_ adsorption on the surface of Ce-La solid solution is consistent with the oxygen vacancy content on the solid solution surface, which proves that the oxygen vacancy on the surface has an important influence on the CO_2_ adsorption [35].

In order to further verify the reason for the change of oxygen vacancies on the solid solution surface, the XPS spectra of Co 2p_3/2_ were analyzed, the results are shown in Figure 5e. Two peaks appear near the binding energy of 778.0 eV and 796.0 eV belong to the characteristic peaks of Co^3+^. A weak peak between 785.0 and 788.0 eV is the characteristic peak of Co^2+^ [36], indicating that Co is mainly dispersed on the solid solution surface in the form of Co_3_O_4_ [37]. The characteristic peak of pure Co_2_O_3_ is at the binding energy of 779.2 eV, while the characteristic peak of CoO is 780.4 eV. When Co is doped on Ce-La solid solution, Co ions are easily enriched on the surface of the catalyst, which is easily oxidized to Co^3+^ during calcination in oxidizing atmosphere, and the increasing of Co^3+^ will lead to a lower binding energy. The binding energy of Co on the XPS spectra of Ce-La-Co shifted from 782.2 eV to 778.8 eV, indicating that Co^2+^ had a tendency to transform into Co^3+^ in a higher valence state [38]. The above results showed that the doping of Co was conducive to the reaction of Ce^4+^ + Co^2+^ → Ce^3+^ + Co^3+^, thereby promoting the formation of oxygen vacancies on the catalyst surface.

### 2.6. Catalytic Activity of the Catalysts for DMC Synthesis

The catalytic performance of the solid solutions was evaluated by synthesis DMC from CH_3_OH and CO_2_, the result as shown in Figure 6. The results showed that the activity of the catalysts increased with doped Co and Zr, and the Co-Ce-La solid solution was the best activity. Under certain reaction condition, the yield of DMC reached 316 mmol/g, which was 52% higher than that of Ce-La solid solution (200 mmol/g). The reason is when Co doped in the CeO_2_ lattice, the electron transfer effect between Co^2+^ and Ce^4+^/Ce^3+^ leads to the change of Ce^3+^ content in the catalyst and affects the oxygen vacancy content on the catalyst surface. The increase of oxygen vacancy content enhances the activation of catalyst for CO_2_, which is conducive to improve catalyst activity. Combined with XRD, BET, and TEM results, the M_x_O_y_ doping on Ce-La solid solution did not significantly affect the crystal structure, texture properties, or morphology of the catalyst. It was believed that the number of acid-base sites and the content of oxygen vacancies on the catalyst surface would significantly affect the reaction activity of CH_3_OH and CO_2_ for the synthesis of DMC [39].

It can be seen from Table 4 that the catalytic activity of Ce-La-M catalyst increased with the increase of Ce^3+^ content on the catalyst surface. Among these catalysts, the Ce-La-Co catalyst with the highest Ce^3+^ concentration on the surface showed the best catalytic performance. When doped metal ions in the CeO_2_ lattice, to maintain the overall electrical neutrality of Ce-La solid solution, more Ce^3+^ was produced, which resulted in an increase content of oxygen vacancies, as shown in Table 4, which revealed the change trend of Ce^3+^ is consistent with oxygen vacancy. According to the previous DFT studies, CO_2_ could be activated at defective oxygen vacancies, but not on a perfect surface [40]. So, the increased content of oxygen vacancies on Ce-La solid solution surface can enhance the adsorption of CO_2_.

It was found that there was a linear relationship between the activity of Ce-La solid solution catalyst and the surface oxygen vacancy content. Research showed that the oxygen vacancy as Lewis basic site on the active surface of the catalyst could promote the adsorption, activation, and dissociation of CO_2_ [41]. CO_2_ as Lewis acid could form a two-tooth adsorption state at the medium-strength basic sites. The two-tooth adsorption state of CO_2_ reacted with CH_3_O* dissociated by methanol to form methyl carbonate active intermediates, while other adsorption modes of CO_2_ such as bicarbonate, single-tooth adsorption state, and bridge adsorption state could not complete the above process [42]. Combined with CO_2_-TPD, when Co and Zr were added, the increase in the number of medium-strength bases on the catalyst surface is conducive to the improvement of catalytic activity. When Mg was added, the number of strong bases gradually increased with the increase in the number of medium-strength bases. The adsorption of CO_2_ at strong bases sites will form a stable CO_3_^2−^ structure, which is difficult to combine with methanol to generate active intermediates, resulting in the decrease of catalyst activity [29]. The acidic sites on the surface of the catalyst mainly act as the activation of methanol molecules. Methanol molecules can be dissociated at the acidic sites to form hydroxyl and methyl and adsorbed on the surface of the catalyst. The latter can react with methyl carbonate intermediates to form DMC [43]. Therefore, the catalytic activity of Co and Zr is better than that of Mg.

According to the experimental results, the reaction mechanism of CO_2_ and CH_3_OH to DMC was speculated, as shown in Figure 7. Taking Co doped Ce-La solid solution as an example. Firstly, part of Ce ion was replaced by Co ion in Ce-La solid solution, then electron transfer between Ce^4+^ and Co^2+^ occurred in the lattice of Co^2+^ doped solid solution, and the concentration of Ce^3+^ increased to maintain the electrical neutrality of the solid solution. Combined with XPS characterization and above research, the higher the content of Ce^3+^ is, the higher surface oxygen vacancies content [44], so Ce-La-Co catalyst formed many oxygen vacancies as Lewis base sites, which CO_2_ molecules could adsorb on the oxygen vacancies of the catalyst by oxygen atoms. Then the methanol molecules are adsorbed on the metal ions adjacent to the oxygen vacancy of the catalyst by oxygen atoms and H_2_O is desorbed, then another methanol molecule is adsorbed on another adjacent metal ion at the oxygen vacancy to form an intermediate. Finally, the intermediate is converted to dimethyl carbonate and releases oxygen vacancies through surface reaction.

Combined with XPS and CO_2_-TPD, Co doping helps to increase the specific surface area of the catalyst, promoting CO_2_ adsorption and enhancing the redox performance, which formed amount of oxygen vacancies on the surface of the catalyst. In the process of catalyst reaction, Co ions with different valence states on the surface of the catalyst, Co^2+^ changes into Co^3+^, resulting in the redox cycle between Ce^4+^ and Ce^3+^. The valence change between ions is conducive to electron transfer, which benefits to increase the proportion of Ce^3+^. The presence of large number of Ce^3+^ helps to generate more oxygen vacancies, while oxygen vacancies are Lewis sites, and more oxygen vacancies are conducive to the adsorption of CO_2_. In addition, a small amount of LaCoO_3_ perovskite structure formed by doping Co on Ce-La also contributes to the formation of oxygen vacancies, which greatly improves the catalytic performance of the catalyst for CO_2_ oxidation.

## 3. Materials and Methods

### 3.1. Materials

Cerium (III) nitrate hexahydrate (99%, AR); methanol (99.5%, AR) 2-cyanopyridine (98%, AR); 1-Butanol (99%, AR) and P123 were purchased from Ron company (Shanghai, China). Cobalt nitrate (99%, AR); lanthanum nitrate (99%, AR); magnesium nitrate (99%, AR); zirconium oxychloride (99%, AR); citric acid (99.5%, AR) were purchased from Feng Chuan Chemical Reagent Co., Ltd. (Tianjin, China). Nitric acid (67%) was purchased from Lasvit Co., Ltd. (Tianjin, China).

### 3.2. Catalyst Preparation

First, 1.36 × 10^−4^ mmol P123, 5 mmol citric acid and 16 mmol HNO_3_ were dissolved in 70 mmol 1-butanol under magnetic stirring, then a certain amount of Ce(NO_3_)_3_·6H_2_O and La(NO_3_)_3_·6H_2_O were added. The molar ratio of Ce and La in the mixture was 9:1. The mixture was stirred for 2 h at room temperature until transparent sol was formed. Volatilize solvent in oven at 393 K for 4 h. The yellow powder was calcined at 823 K for 4 h (heating rate 2 K/min). Finally, Cerium lanthanum solid solution product was obtained, named Ce-La.

Then, 1.36 × 10^−4^ mmol P123, 5 mmol citric acid, and 16 mmol HNO_3_ were dissolved in 70 mmol 1-butanol under magnetic stirring, then a certain amount of Ce(NO_3_)_3_·6H_2_O, La(NO_3_)_3_·6H_2_O, Mg(NO_3_)_2_·4H_2_O, ZrOCl_2_·8H_2_O and Co(NO_3_)_2_·6H_2_O were added. The molar ratio of Ce, La, M (Zr, Co, Mg) in the mixture was 9:1:1, and the molar ratio of metal salt to citric acid was 1:2. The mixture was stirred for 2 h at room temperature until transparent sol was formed. Volatilize solvent in oven at 393 K for 4 h. The yellow powder was calcined at 823 K for 4 h (heating rate 2 K/min). Finally, ternary cerium lanthanum solid solution product was obtained, named Ce-La-M.

### 3.3. Physical Characterization

D8 FOCUS powder X-ray diffractometer test catalyst crystal results, Cu Kα ray, tube voltage 40 kV, tube current 40 mA, sample scanning range 5–90°, scanning rate 6°/min. (German Brook AXS Company, Karlsruhe, German); ASAP 2020 specific surface area and porosity analyzer to determine the specific surface area of the material, pore volume and pore size, test conditions: 363 K, vacuum degassing 8 h, adsorbate N_2_, carrier gas He (Micromeritics company, Norcross, GA, USA); Talos F200S field emission high-resolution transmission electron microscope. Acceleration voltage: 20–200 KV (FEI company, Hillsboro, USA); micromeritics Auto Chem 2920 automatic programmed temperature chemical adsorption instrument (Micromeritics, Norcross, GA, USA); ESCALAB 250xi X-ray photoelectron spectrometer was used to characterize the elemental composition and valence state of the catalyst surface, and the excitation source was Al Kα (1486.6 eV). Vacuum 5 × 10^−7^ Pa, voltage: 15 KV, beam: 15 mA (Thermo Fisher Scientific Company, Waltham, MA, USA).

### 3.4. Catalytic Performance Evaluation

So, 0.1 g catalyst, 15 mL methanol and 5 g 2-cyanopyridine were placed in a high-pressure reactor with PTFE lining. Firstly, 0.5 MPa N_2_ gas was filled into the reactor, and 0.5 MPa N_2_ gas was filled again after emptying. The air in the reactor was blown out of the reactor three times repeatedly. Then, 4 MPa CO_2_ was slowly filled into the reactor, and then the temperature began to rise. When the reaction temperature reached 413 K, the reaction was kept at constant temperature for 5 h. After the reaction, 3 g supernatant was taken, and 0.03 g 1-butanol was dropped into the supernatant as the internal standard. After shaken well, the supernatant was standing, and the internal standard method was used for analysis. In this work, the DMC yield was defined as follows.
$$\mathrm{DMC}\;\mathrm{yield}=\left(\frac{n\;DMC(mmol)}{m\;catalyst(g)}\right)$$

## 4. Conclusions

The citric acid coordinated rare earth metal lanthanum cerium solid solution doped with transition metal Co, alkali metal Mg, and Zr ternary solid solutions was prepared by ligand assisted self-assembly method, and the catalytic performance for direct synthesis of dimethyl carbonate from CH_3_OH and CO_2_ was investigated. The experimental results show that the activity of the catalyst is Ce-La-Co > Ce-La-Zr > Ce-La > Ce-La-Mg. The addition of Co is beneficial to the decrease of the number of weak base sites on the catalyst surface and increase the number of medium-strength base sites, which improve CO_2_ adsorption and the fluidity of lattice oxygen to generate more oxygen vacancies, which promote the redox performance. The reaction mechanism of CO_2_ and CH_3_OH to dimethyl carbonate catalyzed by cerium lanthanum solid solution catalyst was speculated.

## Figures and Tables

**Figure 1 molecules-27-00270-f001:**
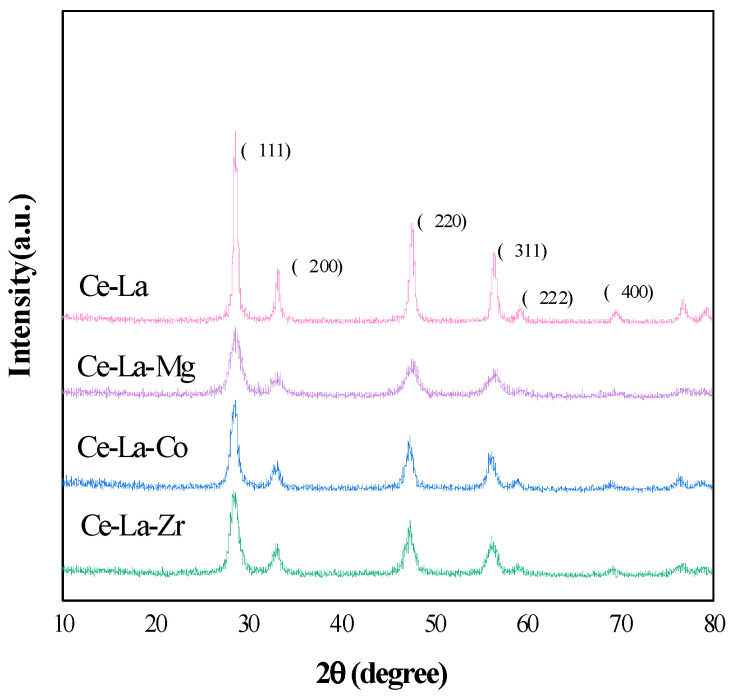
XRD patterns of metal ions doped on cerium lanthanum solid solution.

**Figure 2 molecules-27-00270-f002:**
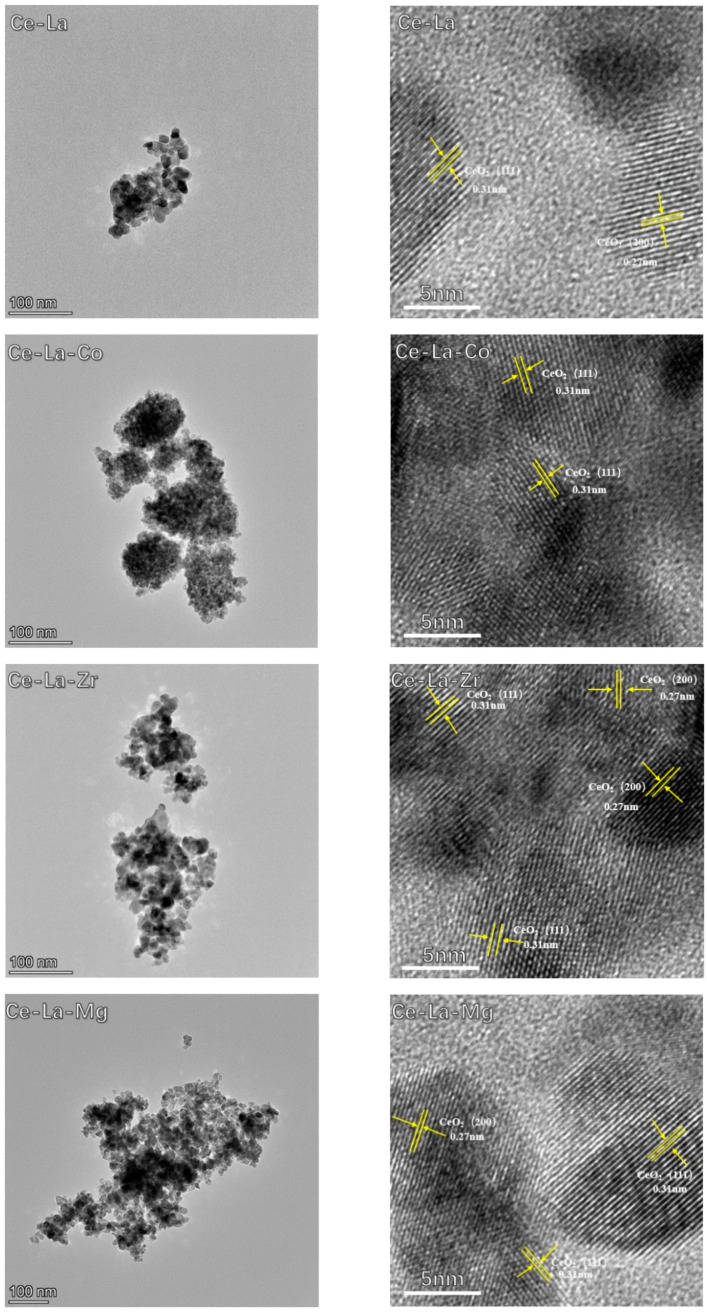
TEM and HRTEM images of metal ion doped cerium lanthanum solid solution.

**Figure 3 molecules-27-00270-f003:**
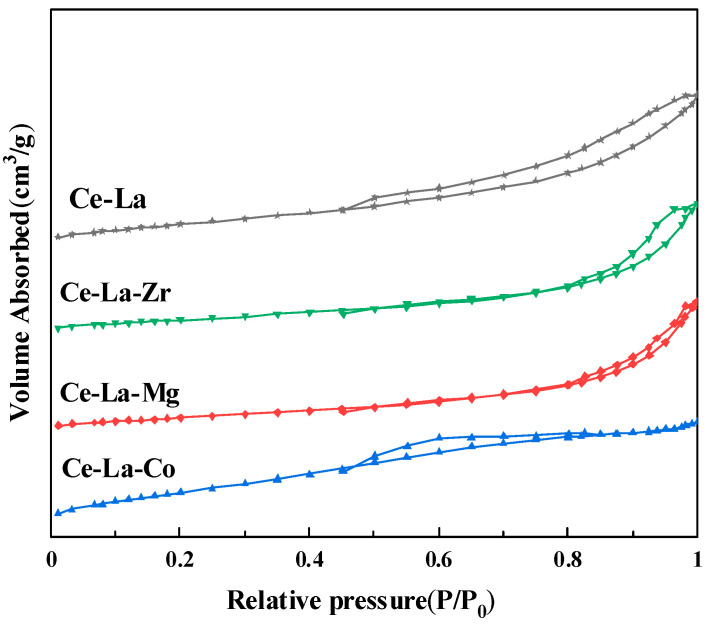
N_2_ adsorption–desorption isotherms of metal elements doped cerium lanthanum solid solution.

**Figure 4 molecules-27-00270-f004:**
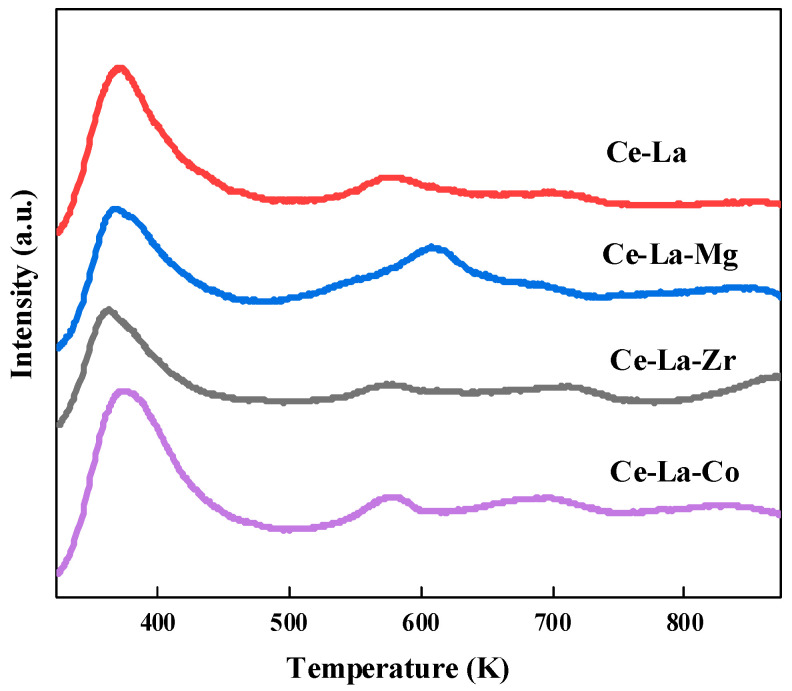
CO_2_⁃TPD profiles of metal elements doped cerium lanthanum solid solution.

**Figure 5 molecules-27-00270-f005:**
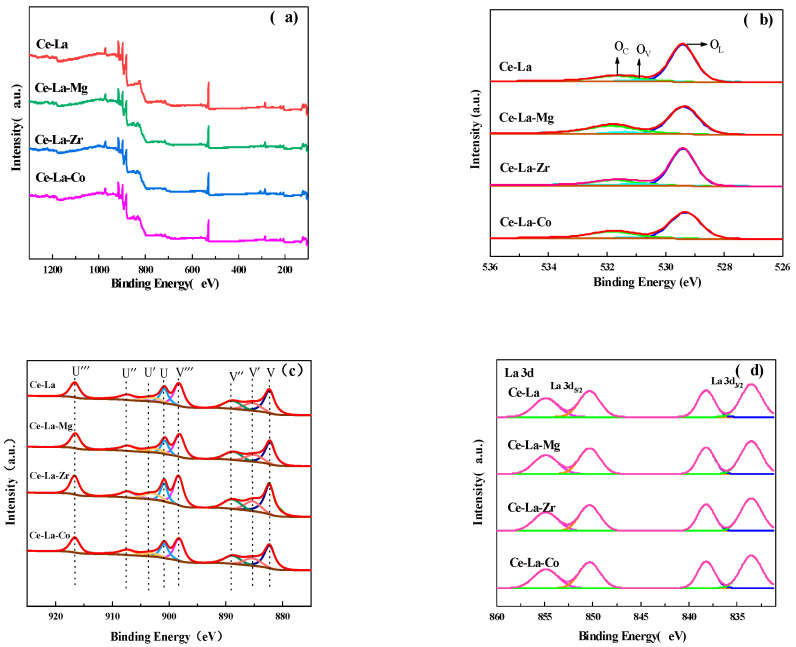
XPS spectra of Ce-La-M solid solution (**a**), O 1s (**b**), Ce 3d (**c**) La 3d (**d**), and Co 2p (**e**).

**Figure 6 molecules-27-00270-f006:**
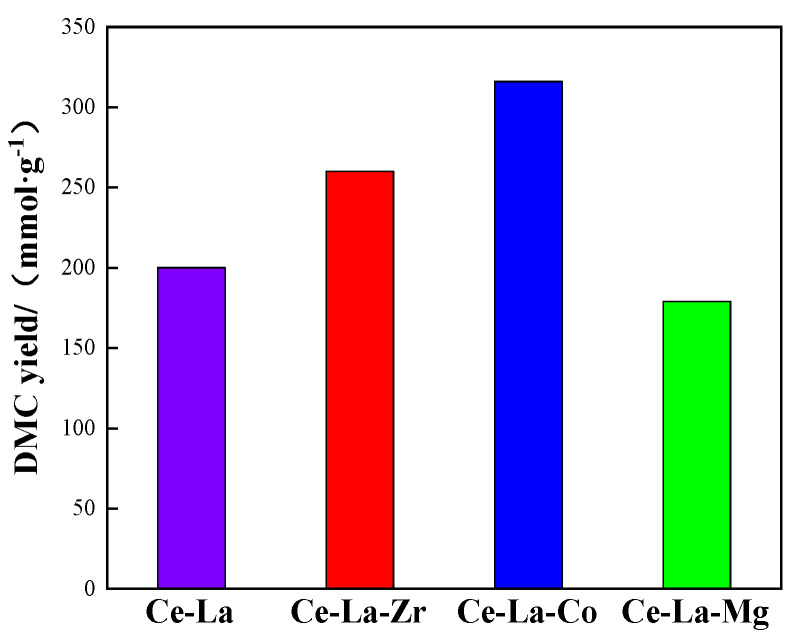
Catalytic performance of the catalyst. Reaction conditions: Ce-La-M 0.1 g, methanol: 2-cyanopyridine = 2:1, Temperature: 413 K, Pressure: 8 MPa.

**Figure 7 molecules-27-00270-f007:**
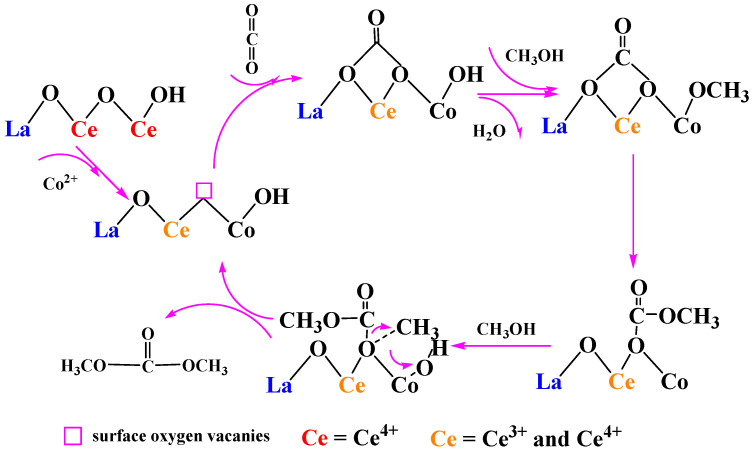
Proposed mechanism of DMC formation from CO_2_ and CH_3_OH at the surface oxygen vacancies over Ce-La-M catalysts.

**Table 1 molecules-27-00270-t001:** Structural and textural properties of Ce-La-M solid solution.

Sample	(111) Plane		Lattice Parameter	Particle Size
	2θ (°)	D (nm)	(nm)	(nm)
Ce-La	28.46	0.3132	0.54255	9.036
Ce-La-Mg	28.49	0.3129	0.54199	7.437
Ce-La-Co	28.56	0.3121	0.54069	8.718
Ce-La-Zr	28.61	0.3116	0.53977	9.111

**Table 2 molecules-27-00270-t002:** Textural parameters of solid solution.

Sample	BET Surface Area/(m^2^/g)	Pore Size/(nm)	Pore Volume/(cm^3^/g)
Ce-La	71	9.1	0.08
Ce-La-Zr	72	9.6	0.11
Ce-La-Mg	74	9.8	0.09
Ce-La-Co	77	9.9	0.13

**Table 3 molecules-27-00270-t003:** Adsorption capacity of CO_2_ of cerium lanthanum solid solution.

Sample	CO_2_ Adsorption (mmol/g)			
	Weak < 473 K	Medium-strength (473~673 K)	Strong > 673 K	Total
Ce-La	0.56	0.06	0.04	0.66
Ce-La-Mg	0.44	0.16	0.15	0.75
Ce-La-Zr	0.45	0.13	0.12	0.70
Ce-La-Co	0.55	0.17	0.07	0.79

**Table 4 molecules-27-00270-t004:** Surface concentrations of Ce and O estimated by XPS.

Sample	Molar Fraction (%)		
	Ce^3+^ (%)	Ce^4+^ (%)	OV (%)
Ce-La	15.32	84.68	9.47
Ce-La-Mg	15.43	84.57	9.94
Ce-La-Zr	15.74	84.26	10.28
Ce-La-Co	16.73	83.27	12.26

## Data Availability

The data presented in this study are available on request from the corresponding author.

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
