# Peer review of "Cobalt Element Effect of Ternary Mesoporous Cerium Lanthanum Solid Solution for the Catalytic Conversion of Methanol and CO2 into Dimethyl Carbonate"

_molecules, 2022, doi:10.3390/molecules27010270_

Round 1
Reviewer 1 Report
The authors reported ternary mesoporous cerium lanthanum solid solution for the catalytic conversion of methanol and CO2 into dimethyl carbonate. However, there are some issues in the article. I suggest that it be accepted after minor modifications:
- Why do you choose the doping of Mg, Co and Zr for comparison? How is the doping concentration determined? Why the molar ratio of Ce, La, M (Zr, Co, Mg) in the mixture was 9:1:1?
- In XPS spectral analysis, why is the high resolution XPS of La not provided? Co is also a metal element with multiple valence states, Co in Fig. 5d needs to be divided into bivalent and trivalent peaks, and corresponding analysis and citation should be carried out.
- The redundant figures 6b and c should be removed, the relevant data have been listed in Table 4.
- What is the stability of the catalyst? Do the structure and properties of the catalyst change before and after the reaction?
Reviewer 2 Report
The paper of Xu Li and co-authors is an interesting fundamental work on synthesis solid solutions (Ce-La-M, M = Co, Zr, Mg) and testing it in the catalysis of dimethyl carbonate formation. It was shown that Co-doped material has the highest potential as it more effective than starting Ce-La one. The current work seems interesting to me and I recommend it to publish in Molecules.
But I have some questions:
Authors thought that Ce4+ oxidize Co2+ to form Ce3+ which ionic radii is bigger and it leads to more surface oxygen vacancies. So how can authors explain the formation of Ce3+ in solid solutions with redox-inactive Mg2+ and Zr4+?
The molar ratio of doped metal is 9-1-1 (Ce-La-M), but there are several sentences in manuscript which are not clear:
…doped metal elements on Ce-La solid solution did not significantly affect the particle size and particle size distribution
… M doping does not significantly affect the texture properties of the Ce-La solid solution
… texture properties of catalysts changed little by doping metal elements
… no obvious peaks 180 of doped metal elements (Zr, La, Mg, Co) are observed
How can we be sure that we have doped material instead of mixture of substances? And why authors didn’t make ICP-MS analysis?
